# Learning to remember: Dynamic Generative Memory for Continual Learning

## Abstract

Continuously trainable models should be able to learn from a stream of data over an undefined period of time. This becomes even more difficult in a strictly incremental context, where data access to previously seen categories is not possible. To that end, we propose making use of a conditional generative adversarial model where the generator is used as a memory module through neural masking to emulate neural plasticity in the human brain. This memory module is further associated with a dynamic capacity expansion mechanism. Taken together, this method facilitates a resource efficient capacity adaption to accommodate new tasks, while retaining previously attained knowledge. The proposed approach outperforms state-of-the-art algorithms on publicly available datasets, overcoming catastrophic forgetting.

## 1 Introduction

Continual lifelong learning is unquestionably one of the most important cognitive skills of humans and artificial neural networks alike. This can be largely attributed to the fact that every time something new is learned, a little of bit of old knowledge gets overwritten. Therefore, addressing this is of particular relevance in a continual learning setting where forgetting is largely unavoidable. In this regard, this phenomenon in context of artificial neural network algorithms is commonly referred to as catastrophic forgetting (McCloskey & Cohen, 1989; Ratcliff, 1990; French, 1999).

Catastrophic forgetting is believed to occur due to a lack of a plastic component in the neuron connections (Kirkpatrick et al., 2016; Zenke et al., 2017). In analogy with biological systems, neural or synaptic plasticity is the capability of a neuron or synapse to "lock" itself to a state or a connection, thus retaining previously attained knowledge. Indeed, it has been shown that this plasticity is responsible for maintaining previously acquired structure in the neo-cortical circuits of brains (Cichon & Gan, 2015; Hayashi-Takagi et al., 2015).

Several recent approaches try to mitigate forgetting by simulating neuroplasticity in artificial neural networks. To that end, one possibility is to identify critically important parameters of a network w.r.t. a given task, and imposing a penalty for changing them for learning another task (Kirkpatrick et al., 2016; Chaudhry et al., 2018; Zenke et al., 2017; Aljundi et al., 2017). Common to all these methods is that they seek to find a point in the model parameter space that minimizes the loss jointly for all previously learned tasks. However, the existence of such point is not always guaranteed, hence the hard attention to the task (HAT) mechanism was proposed Serrà et al. (2018). HAT finds a parameter subspace for each task that can overlap with other tasks' parameter subspaces. The optimal solution is then found in the corresponding parameter subspace of each task. These solutions can be seen as reserving sub-spaces of the neural network parameter space for each task at hand and driving the subsequent network updates into the free-capacity space.

Another important factor in the continual learning setting is the ability to scale, i.e. to maintain sufficient capacity to accommodate for a continuously growing number of tasks. Given equitable resource constraints, it is inevitable that with a growing number of tasks to learn, the model capacity is depleted at one point in time. Without a smart and adaptive capacity strategy network expansion can swiftly go beyond existing hardware limits, or unnecessarily waste resources by reservation.

In order to address catastrophic forgetting in a continual learning setting some approaches rely on storing raw samples of previously seen data and making use of replay strategies during the training

of subsequent tasks. However, this starkly contrasts with the natural learning mechanisms of the brain, which does not feature the retrieval of raw information identical to originally exposed impressions Mayford et al. (2012). What is more, storing raw samples of previous data may violate data privacy and memory restrictions in the real world applications. Instead of storing raw samples, the proposed approach relies upon on the idea of training a deep generative model to memorize previously seen data distributions. Simultaneous with the adversarial training on each incoming data batch, we learn a sparse mask for the layer activations of the generator network. This serves the purpose of encouraging parameter re-usability across tasks. The values of the learned mask correspond to the plasticity of the connections between the layers. Thus units with low plasticity are reserved for previous tasks and can be reused but not changed during the subsequent training.

Our contribution is twofold: **(a)** we address the catastrophic forgetting problem in continual learning; we introduce **Dynamic Generative Memory** (DGM) - an adversarially trainable generative network endowed with neuronal plasticity through efficient learning of a sparse parameter attention mask.[1] for layer activations; a *single* generator is able to incrementally learn new information during normal adversarial training *without the need to replay previous knowledge*; **(b)**W e address the scalability problem in continual learning proposing an **adaptive network expansion mechanism**, facilitating resource efficient continual learning. We compare the proposed method to state-of-the-art approaches for continual learning.

## 2 RELATED WORK

Among the first works dealing with catastrophic forgetting in the context of lifelong learning are (French, 1999; McCloskey & Cohen, 1989; Ratcliff, 1990). In contrast to the proposed approach, these methods tackle this problem by employing shallow neural networks, whereas our method makes use of modern deep architectures. In this regard, lately a wealth of works dealing with catastrophic forgetting in context of deep neural networks have appeared in the literature, see e.g., (Kirkpatrick et al., 2017; Zenke et al., 2017; Li & Hoiem, 2016; Serrà et al., 2018; Aljundi et al., 2017). However, all of these methods have been proposed for a "task-incremental learning" setup, where a sequence of disjoint tasks should to be learned one after the other by a single network. In our work, however, a completely different incremental methodology is used. We specifically propose a method to overcome catastrophic forgetting within the "class-incremental learning" setup. The key difference in the task-incremental setup is that the model learns a separate classifier for each task (i.e., multi-head), whereas in the latter the model learns only a single classifier for all of the observed classes of all tasks (i.e., single-head). Several continuous learning approaches (Rebuffi et al., 2016; Nguyen et al., 2017; Kemker & Kanan, 2017), address catastrophic forgetting in the class-incremental setting, i.e. by storing raw samples of previously seen data and making use of them during the training of subsequent tasks.

There is a growing interest in employing deep generative models for memorizing previously seen data distributions instead of storing old samples. Thus Shin et al. (2017); Wu et al. (2018) rely on the idea of generative replay, which requires retraining the generator from scratch at each time step on a mixture of synthesized images of previous classes and new real samples. Apart from being inefficient for training, it is severely prone to "semantic drifting". Namely, the quality of images generated at every memory replay point highly depends on the images generated during previous replays, which can result in loss of quality over time. In contrast to the methods described above, we propose to utilize a single generator that is able to incrementally learn new information during the normal adversarial training without the need to replay previous knowledge. We achieve it though efficiently learning a sparse mask for the layer activations of the generator network.

Similar to our method, Seff et al. (2017) proposed to avoid retraining the generator at every timestep on the previous classes, by applying per-parameter regularization selectively in the generative network. We pursue a similar goal with the key difference that we expand the capacity of generator dynamically (with increasing amount of attained knowledge) instead of regularizing parameters of previous tasks. A similar strategy has been proposed in Yoon et al. (2018). However, they propose to keep track of the *semantic drift* in every neuron and then expand the network by duplicating neurons that are subject to sharp changes. In contrast, we compute weights importance concurrently during the course of network training by modeling the neuron behavior using an explicit binary mask. As

---

[1]Represented as a learnable binary mask applied to the network's activations.

a result, our method explicitly does not require any further network retraining after adding new capacity. Another approach relying on dynamic network extension was proposed by Rusu et al. (2016) and improved by Schwarz et al. (2018). In Progressive Neural Networks Rusu et al. (2016) authors propose to assign a separate column (network) to every new task, while insure positive knowledge transfer by maintaining trainable connections from the pretrained columns of previous task to the current task's column. Schwarz et al. (2018) address the scalability problem of Progressive Networks by only keeping two columns: a knowledge base and an active column while distilling the knowledge continuously from the later to the former one. In our approach we only keep a single column for all tasks in which knowledge transfer happens naturally, since parameters of old tasks can be freely reused by the new ones.

Other approaches like (Kemker & Kanan, 2017; Kamra et al., 2017) try to explicitly model short and long term memory with separate networks while transferring the knowledge from the former one to the later in a separate "sleeping" phase. In contrast to these methods our approach does not explicitly keep two separate memory locations, but models it implicitly in a single memory network. The memory transfer occurs during the binary mask learning from non-binary (short term) to completely binary (long term) values.

Most related to our paper is the work by Serrà et al. (2018), which considers a task-based hard attention mechanism that preserves previous tasks' information without affecting the current task's learning. In this regard, they propose to learn a binary ask for the parameters of a base network similar to Courbariaux et al. (2015) and Mallya & Lazebnik (2018) and Mancini et al. (2018). In this fashion, while the parameters of the network are kept constant, the binary mask is used to find a task specific perturbation of the base network that maximizes performance on incrementally introduced new tasks. We build upon this idea and extend it by a mask for the neurons to be jointly learned with the network parameters directly within the generator, thus allowing it to work in class-incremental setup without storing samples of previous tasks. Such binary mask also provides us with an efficient growth mechanism for network capacity by explicit modeling of plasticity and abandoning of over-capacity.

## 3 PRELIMINARIES

Adopting the notation of Chaudhry et al. (2018), let $D_k = \{(x_i^k, y_i^k)\}_{i=1}^{n^k}$ denote a collection of data belonging to the task $k \in \mathcal{K}$, where $x_i^k \in \mathcal{X}$ is the input data and $y_i^k \in y^k$ are the ground truth labels.

While in the (standard) non-incremental multi-task setup the entire data $D = \cup_{k=0}^{|K|} D_k$ is available at once, in a task-incremental setup the incoming dataset $D_k$ becomes available to the model continuously and specifically only during the learning of task $k$. Thereby, $D_k$ can be composed of a collection of class items from the same task, or more generally only from a single class. We address the latter class-incremental setup, in which the model is presented data from a single class at a time. Furthermore, during test time the output space covers all the labels observed, independently from the task they belong to: $\mathcal{Y}^k = \cup_{j=1}^k y^j$. Such an evaluation scenario is often referred to as a single-head evaluation (Chaudhry et al., 2018; Rebuffi et al., 2016) and is predominantly used in class-incremental learning.

## 4 METHOD

In the continuous learning setup the task at a time $t$ is to learn the parameters $\theta$ of the predictive model $f_\theta : \mathcal{X} \rightarrow \mathcal{Y}^t$, where $\mathcal{Y}^t = \cup_{j=1}^t y^j$ denote all classes seen so far. This requires solving the following problem:

$$\min_\theta \mathcal{L}(\theta, D_t) + \Omega(\theta), \qquad (1)$$

where $t$ is the task index, $\mathcal{L}$ is the loss function and $\Omega(\theta)$ is the regularization term. It should be noted that if $D_t = \{(x^k, y^t)\}$, i.e. the training data of task $t$ only contains a single class, e.g. $|\{y\}| = 1$, the loss can not be computed without referring data from previously seen classes. However, storing any training data violates the strictly incremental setup. In order to overcome this,

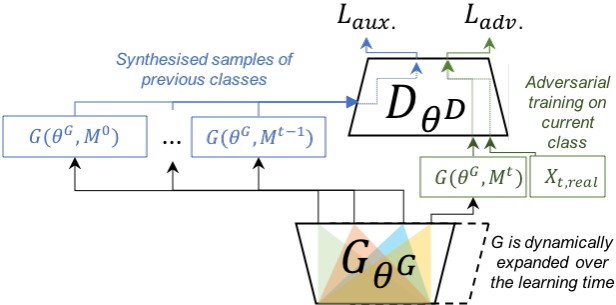

Figure 1: Dynamic generative memory: The AC-GAN Odena et al. (2016) architecture allows simultaneous adversarial and auxiliary training, in which auxiliary output is trained on the real samples of the current task and synthesized sample of previously seen tasks. During the adversarial training with real and fake samples of current class a connection plasticity in the generator is learned simulataniously with the weights of the base network. Connections plasticity is represented by task specific, trainable binary masks for base netwrok's activations.

a generator $G(y_t, z, m^t)$ based on AC-GAN (Odena et al., 2016) is employed to synthesize samples of the previously seen classes.

However, in order to facilitate learning in an incremental setting and to mitigate catastrophic forgetting, the generator $G$ is further associated with: (i) a learned binary mask - see Sec. 4.1, (ii) a dynamic capacity expansion mechanism - see Sec. 4.2.

The learned binary masks models the plasticity of neurons and thus avoids overwriting important units by directing SGD updates in segments of the neural network that correspond to free capacity, or capacity that can be shared. The dynamic network expansion facilitates adding neurons in order to provide sufficient capacity to deal with new tasks, capacity that is efficiently allocated based on task-wise mask allocation.

To this end, in every SGD step $t$ we learn a binary mask $M^t = [m_1^t, ..., m_l^t]$ based on the previous layer's activations of the generator $G(y_t, z, m^t)$. The binary mask $M_t$ is then modulated with the layer activation, yielding e.g.

$$y_l^t = m_l^t \odot (W_l \cdot x), \tag{2}$$

for a fully connected layer $l$, where $m_l^t$ is an $n$-element vector and $W_l$ is the weight matrix between layer $l$ and $l-1$ shaped as $m \times n$.

Subsequently, a network $H_\Theta$ serves both as a discriminator for newly synthesised samples and classifier for the overall learning problem. $H_\Theta$ is reinitialized for every task $t$ and retrained using synthesized samples of the previously seen tasks.

## 4.1 LEARNING A BINARY MASK

Similarly to other approaches that aim to learn a binary mask for the network parameters (Courbariaux et al., 2015; Serrà et al., 2018; Mallya & Lazebnik, 2018), we make use of real valued mask embeddings $e_l^t$, which are scaled by a positive scaling parameter $s$, and passed through a threshold function $\sigma(x) \in [0, 1]$. The binary mask is thus given by $m_l^t = \sigma(se_l^t)$

We use sigmoid function as a pseudo step-function such that a gradient flow to train mask embeddings $e$ is ensured. As pointed out by Serrà et al. (2018), the scaling parameter $s$ controls the degree of binarization of the mask $m$. Specifically, the higher the value of $s$, the higher the extent of binarization, and vise versa, i.e. $m_l^t \to \{0, 1\}$ for $s \to \infty$, $m_l^t \to 0.5$ for $s \to 0$. Over the course of training, $s$ is incrementally annealed from 0 to $s_{max}$, largely following the scheme proposed by Serrà et al. (2018):

$$s = \frac{1}{s_{max}} + (s_{max} - \frac{1}{s_{max}})\frac{i-1}{I-1}, \tag{3}$$

where $i$ corresponds to the current epoch and $I$ is the total number of epochs.

In order to prevent the overwriting of the knowledge related to previously seen classes in the generator network, the gradients $g_l$ of the weights of each layer $l$ are multiplied by the reverse of the cummulated binary masks for all previous tasks:

$$g_l' = [1 - m_{l,m \times n}^{\leq t}]g_l, \tag{4}$$

$$m_l^{\leq t} = \max(m_l^t, m_l^{t-1}), \tag{5}$$

where $g_l'$ corresponds to the new weights gradient and $m_{l,m \times n}^{\leq t}$ is a cummulated mask expanded to the shape of $g_l$ ($n$ times duplication of $m^{\leq t}$ to match sizes).

In the cummulated attention mask the parameters that are important for the previously learned classes are masked with values of "1" (or close to 1), whereas free parameters that can be modified during further training are masked with "0". Parameters reserved for previous tasks (marked with "1") can be used during further training, however, their values can not be modified. The higher the sparsity of the mask, the higher the number of parameters that can be modified during consecutive training steps.

Similarly to Serrà et al. (2018), we promote sparsity of the binary mask by adding a regularization term $R^t$ to the loss function $L_G$ of the AC-GAN generator:

$$R^t(M^t, M^{t-1}) = \frac{\sum_{l=1}^{L-1} \sum_{i=1}^{N_l} m_{l,i}^t (1 - m_{l,i}^{<t})}{\sum_{l=1}^{L-1} \sum_{i=1}^{N_l} 1 - m_{l,i}^{<t}}, \tag{6}$$

where $N_t$ is the number of parameters of the layer $l$. Parameters that were reserved previously by other tasks are not regularized, while parameters that have never been used are. This promotes reusing neurons over reserving new ones and promotes efficiency in neuron allocation per task.

**Joint training:** The system must learn to perform three tasks: A generative, a discriminative and finally a classification task in the strictly incremental class setup.

As such, using task labels as conditions, the generator network (G) must learn from a training set $X_t = \{X_1^t, ..., X_N^t\}$ to generate images for task $t$. AC-GAN's conditional generator synthesizes images $x_t = G_{\theta_G^t}(t, z, M_t)$, where $\theta_G^t$ represents the parameters of the generator network at task $t$, $z$ a random noise vector, and $M_t$ the computed binarization mask for task $t$. The discriminator network is used to perform two tasks. First, a discriminative task, determining whether sample $x_f$ is real or fake. Second, a classification task, indicating whether $x_f$ can be labelled as part of the task $t$. To achieve both, the final layer of the network has two branches corresponding to each task, and the base network added to each layer is parameterized with $\theta_H^t$ and $\theta_C^t$ respectively. The parameters corresponding to the three tasks, $\theta_G^t, \theta_H^t, \theta_C^t$ are optimized in an alternating fashion. As such, the generator optimization problem can be seen as minimizing $\mathcal{L}_\mathcal{G} = \mathcal{L}_s^t - \mathcal{L}_c^t + \lambda_{RU} R^t$, with $\mathcal{L}_c$ a classification error on the auxiliary output, $\mathcal{L}_s$ a discriminative loss function used on the binary output layer of the network, and $\lambda_{RU} R^t$ the regularizer term expanded upon in Equation 6.

To promote efficient neuron use taking into consideration the amount of the network already in use, the regularization weight $\lambda$ is multiplied by the ratio $\alpha = \frac{S_t}{S_{free}}$, where $S_t$ is the size of the network before training the task $t$, and $S_{free}$ is the number of free neurons. This ensures that less parameters are reused during the beginning stages of training, and more during the later stages.

The discriminator is optimized similarly through minimizing $\mathcal{L}_\mathcal{D} = \mathcal{L}_c^t - \mathcal{L}_s^t + \lambda_{GP} \mathcal{L}_{gp}^t$, where $\mathcal{L}_{gp}^t$ represents a gradient penalty term implemented as in Gulrajani et al. (2017), to ensure a more stable training process.

## 4.2 DYNAMIC NETWORK EXPANSION

As discussed by Mancini et al. (2018) the more significant the domain shift between the learned data batches, the quicker the network capacity will be exhausted and the effects related to catastrophic forgetting will manifest. This can be attributed to the overall decline in sparsity of the of the accumulated mask $m_l^{\leq t}$ over the course of training. In order to avoid this effect, we keep the sparsity of the binary mask equal for each training cycle $t$.

Assuming a network layer $l$ with input vector of size $m$ and output vector of size $n$. At the beginning of the initial training cycle on $D_0$ the binary mask $m_l$ is initialized at size $n$, with zero sparsity. Thus, all neurons of the layer will be used, with all $n$ values of the binary mask set to 0.5 (real-valued embeddings $e$ are initialized with 0).

After the initial training cycle with the sparsity regularizer $R^0$, the sparsity of the mask will decrease to $n - \delta_t$, where $\delta_t$ is the number of parameters reserved for the generation task $t$. Subsequently, after the training cycle $D_0$, we expand the number of output units $n$ of the layer $l$ by $\delta_t/m$. This guarantees that the free capacity of the layer is kept constant at $n$ for each learning cycle.

## 5 EXPERIMENTAL RESULTS

In the following section we provide a qualitative and quantitative evaluation of our method on a number of publicly available datasets. Furthermore, we provide a discussion upon the performance of the different components of our system.

### 5.1 EXPERIMENTS

We perform experiments measuring the classification accuracy of our system in a strictly class incremental setup on three benchmark datasets: MNIST (LeCun, 1998), SVHN (Netzer et al., 2011) and CIFAR-10 (Krizhevsky et al., 2014). Both a 10-class and 5-class accuracy is reported, to provide a high level impression on the criticality of the catastrophic forgetting problem: ideally a system that does not forget will see results that are stable, or even increasing as it learns to generalize better from previously seen tasks.

All datasets are used to train a classification network in the strictly incremental setup, and the performance of our method is evaluated quantitatively through comparison with benchmark methods. Note that we compare to both type of methods; (i) where the strictly incremental constraints are relaxed, Rebuffi et al. (2016), Kirkpatrick et al. (2016), Chaudhry et al. (2018), Zenke et al. (2017), (ii) methods that strictly adhere to the setup, making use of a type of memory: Seff et al. (2017), Shin et al. (2017), Wu et al. (2018). Further, we present a qualitative evaluation of CIFAR-10 using a perceptual metric (Heusel et al., 2017).

**Datasets:** The MNIST and SVHN datasets are composed of 60000 and 99289 images respectively, containing digits. The main difference is in the complexity and variance of the data used. SVHN's images are cropped photos containing house numbers and as such present varying viewpoints, illuminations, etc. All images are further resized to 32 x 32 before use. Finally, CIFAR10 contains 60000 $32 \times 32$ labelled images, split in 10 classes, roughly 6k images per class.

**Architectures:** We make use of the same architecture for the MNIST and SVHN experiments, a 3-layer DCGAN (Radford et al. (2015)), with the generator's number of parameters modified to be proportionally smaller than the architecture reported in Wu et al. (2018). The projection and reshape operation is further performed with a convolutional layer instead of a fully connected one. For the CIFAR-10 experiments, we use the CIFAR-10 ResNet architecture proposed by Radford et al. (2015). Note that all architectures used have been modified to function as an AC-GAN, i.e. the discriminator has both a discriminative and a classification role.

### 5.2 RESULTS

A quantitative comparison of the proposed DGM approach with other methods that operate in a single-head regime on the MNIST dataset is listed in Table 1. The first set of methods do not adhere to the strictly incremental setup, and thus make use of stored samples. As such these methods are not generative, and have difficulties in generalizing across different tasks due to inability to capture sufficient diversity. They are sharply outperformed by our method, despite of having access to previously observed data during training.

The second set of methods we compare with use a single-head evaluation protocol, and beyond that do not store any additional data. Our method significantly outperforms Seff et al. (2017) and Shin et al. (2017) on the MNIST benchmark through the integration of the memory learning mechanism directly into the generator network, and the expansion of said network as it saturates to accommodate

| Method | MNIST | | SVHN | | CIFAR10 | |
|---|---|---|---|---|---|---|
| | $A_{10}(\%)$ | $A_5(\%)$ | $A_{10}(\%)$ | $A_5(\%)$ | $A_{10}(\%)$ | $A_5(\%)$ |
| iCarl-S (Rebuffi et al., 2016) | - | 55.8 | - | - | - | - |
| EWC-S(Kirkpatrick et al., 2016) | - | 79.7 | - | - | - | - |
| RWalk-S(Chaudhry et al., 2018) | - | 82.5 | - | - | - | - |
| PI-S (Zenke et al., 2017) | - | 78.7 | - | - | - | - |
| EWC-M (Seff et al., 2017) | 77.03 | 70.62 | 33.02 | 39.84 | - | - |
| DGR-M (Shin et al., 2017) | 85.4 | 90.39 | 47.28 | 61.29 | - | - |
| MeRGAN-M (Wu et al., 2018) | 97.0 | 98.19 | 66.78 | 80.90 | - | - |
| Joint Training | 98.1 | 97.66 | 84.82 | 85.30 | 64.2 | 82.2 |
| DGM (ours) | **99.17** | **98.14** | **68.36** | **84.18** | **50.8** | **62.5** |

Table 1: Comparison to the benchmark presented by [15] of approaches that make use of generative memory. Joint training (JT) represents the the performance of the discriminator trained in non-incremental fashion. Performance of DGM is presented after incremental learning of 5 and 10 classes for MNIST, SVHN and CIFAR10 datasets.

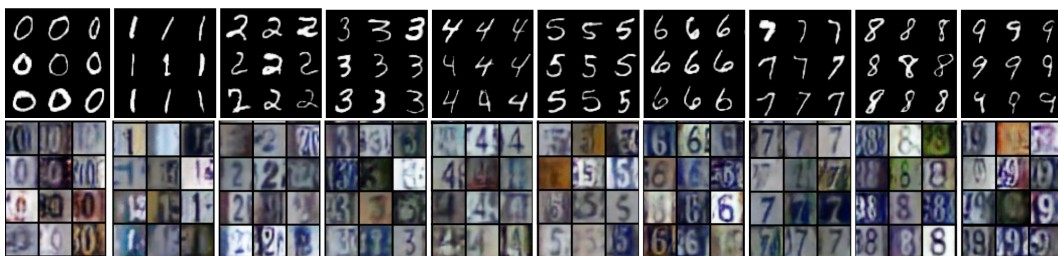

Figure 2: Images generated from MNIST and SVHN after learning ten tasks

new tasks. We yield an increase in performance over Wu et al. (2018) , a method that is based on a similar architecture but does not provide dynamic network expansion and what is more makes use of a replay strategy for memory purposes, losing any information not sampled during training. As it can be observed for both our method and *MeRGAN*, the accuracy reported between 5-tasks and 10-tasks has changed little, suggesting that for this dataset and evaluation methodology both methods have largely curbed the effects of catastrophic forgetting.

Interestingly, *DGM* outperforms joint training on the MNIST dataset using the same architecture. This suggests that the strictly incremental training methodology indeed forced the network to learn better generalizations compared to what it would learn given all the data.

Given the high accuracies reached on the MNIST dataset largely give rise to questions about saturation. We thus perform further evaluation on the more visually diverse SVHN dataset (Netzer et al. (2011)), where increased data diversity translates to a more difficult generation and susceptibility to catastrophic forgetting. In fact, as can be seen in Table 1, the difference between 5-task and 10-task accuracies is significantly larger in all methods than what can be observed in the MNIST experiments.

Our system sharply outperforms all other methods on this task. This can be attributed to a network that is initialized with a small size preventing overfitting on single tasks. At the same time, the network will expand at minimal capacity requirement in order to accommodate joint distribution expansion by integrating the next tasks w.r.t. to all previously seen classes. DGM thus becomes more stable in the face of catastrophic forgetting.

The quality of the generated images after 10 stages of incremental training for both MNIST and SVHN can be observed in Fig. 2. The generative network is able to provide an informative and diverse set of samples for all previously seen classes without catastrophic forgetting.

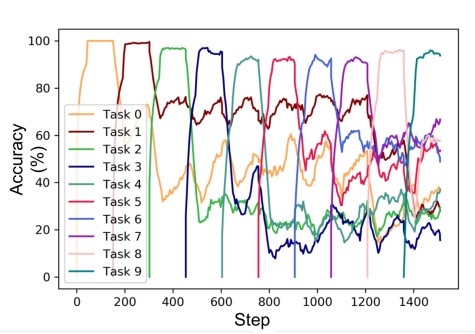 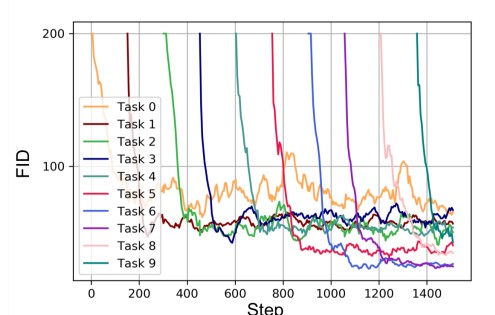

Figure 3: Evolution of the classification accuracy over the 10 tasks of the CIFAR10 dataset. Performance of the currently learned task is always high (real samples are available), it drops during the subsequent tasks due to increasing problem complexity.

Figure 4: Evolution of Frechet Inception Distance (FID), the lower the better. FID decreases during the absence for forgetting is evidenced by the fact that FID does not increase during the learning subsequent task.

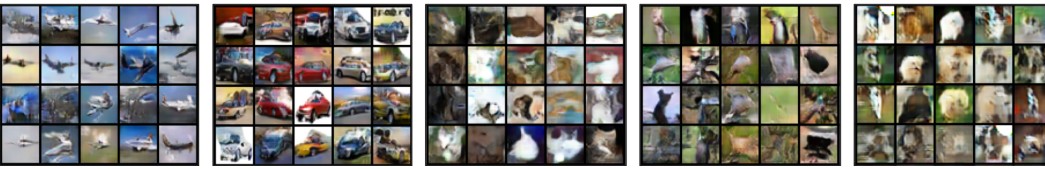

Figure 5: Samples generated for CIFAR10 after 5th task

Finally, on the CIFAR-10 dataset we provide a Frechet Inception Distance (Heusel et al., 2017) metric of the generated images over the first 5 classes in order to asses the perceptual quality of the generation. Figure 4 provides the dynamics of this metric while training across the different tasks. In conjunction with the qualitative results shown in Fig. 5 it can be observed that little to no quality is lost for generating historical task samples. For each newly learned task the discriminator network is reinitialized, but the knowledge contained in the generator is quickly leveraged thus making re-learning a fast process. With the more complex CIFAR samples, more of the task is forgotten, but the task knowledge is maintained as observed in the qualitative results.

## 5.3 DISCUSSION

The primary strength of our proposed method is an efficient network expansion component, whose performance is directly related to how neurons are reserved in the strictly incremental setup.

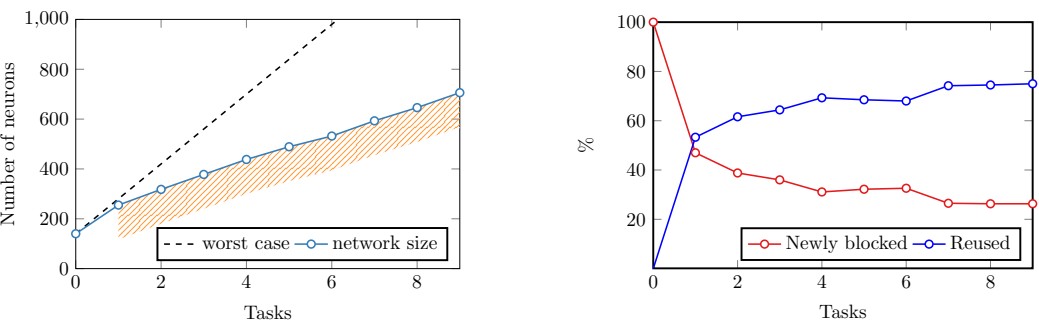

Figure 6: Neural capacity analysis

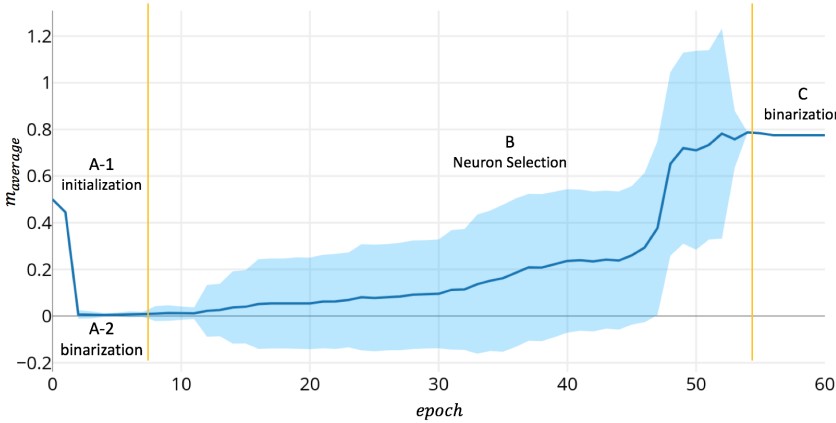

Figure 7: Neuron plasticity during the training of a task $t$. The standard deviation of the mask values is shown as a blue shadow.

We first analyze how learning is accomplished within a given task $t$, and how this further affects the wider algorithm. For a given task $t$, its corresponding binary mask $M_t$ is initialized with the scaling parameter $s = 0$. As observed in Fig. 7 (A-1), at task initialization the mask is completely non-binary, with no new neurons set aside to learn the new task.

As training progresses for the given task, both the scaling parameter $s$ and regularization scaling parameter $\lambda$ are annealed and increased. With heavy regularization and increasing binarization, the network is therefore encouraged to make use of the smallest possible number of neurons, behaviour observed in region (A-2).

But with most mask values near 0, the network's capacity to learn is greatly curtailed. The optimization process pushes the mask to become less binary so that neurons can have better mobility, and a short-term memory is formed. The number of mask values corresponding to useful neurons is steadily increased with the binarization bias, a trend observed in region (B) as the standard deviation of the mask values corresponding to each neuron. This behaviour can be seen as a state transition where the most representative neurons are selected and reserved by the network with neurons that have not made this transition being essentially unused for learning task $t$.

Consequently, neurons can be broadly divided into three types for a given task $t$: (i) neurons that are not used at all (U), which can be seen as a network's free capacity, (ii) neurons that are newly blocked (NB), (iii) neurons that have been reused from previous tasks (R).

Figure 6 presents the evolution of the ratio of the (NB) and (R) types over the total number of neurons. Of particular importance is that the ratio of reused neurons is increasing between tasks, while the ratio of newly blocked neurons is decreasing. These trends can be justified by the network learning to generalize better, leading to a more efficient capacity allocation for new tasks.

An issue with reserving the representative neurons of each task is that the network will eventually run out of capacity - and as such we maintain a constant number $N_{free}$ of unused neurons (Figure 6, shaded area). This is achieved by expanding the generation network after each task with the number of newly blocked neurons denoted as $NB(t)$. Network grow is thus directly linked to how efficiently neurons can be reserved per task, which ultimately depends on how well the generative network can generalize from previously learned tasks. We can observe the efficiency of our network growth method compared to a worst case scenario, where for every task we add the same number of parameters $N_{init}$ to the network.

Indeed, the manner in which $G$ is learned can be seen as learning for both representative and discriminative features. This learning regime has the capacity of generalizing better compared to a network that is simply trained with the entirety of the dataset, as seen in Table 1, where our method outperforms the multi-label baseline on MNIST.

## 6 CONCLUSION

In this work we study the continual learning problem in a single-head, strictly incremental context. Our results suggest that the proposed approach successfully overcomes catastrophic forgetting scenarios by making use of a conditional generative adversarial model where the generator is used as a memory module through neural masking. Furthermore, we show that the proposed dynamic memory expansion mechanism facilitates a resource efficient adaption in order to successfully accommodate learning new tasks. For future work we plan to expand plasticity to also incorporate weights as well as to refine the dynamic capacity expansion.

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
