# OpenReview forum: "Learning to remember: Dynamic Generative Memory for Continual Learning"
_ICLR.cc/2019/Conference_

### Official Review · AnonReviewer2 · 2018-10-27
**Good work on how to prioritize the use of neurons in memory**

**Rating:** 8
**Confidence:** 5

**Review:**

As a paper on how to prioritize the use of neurons in a memory this is an excellent paper with important results.

I am confused by the second part of the paper an attached GAN of unlimited size. It may start out small but there is nothing to limit its size over increased learning. It seems to me in the end it becomes the dominate structure. You start the abstract with "able to learn from a stream of data over an undefined period of time". I think it would be an improvement if you can move this from an undefined time/memory size to a limited size for the GAN and then see how far that takes you.

---

### Official Review · AnonReviewer1 · 2018-10-30
**The expanded generator will also raise the storing problem as that in episodic memory strategy**

**Rating:** 3
**Confidence:** 5

**Review:**

This paper attempts to mitigate catastrophic problem in continual learning. Different from the previous works where episodic memory is used, this work adopts the generative replay strategy and improve the work in (Serra et al., 2018) by extending the output neurons of generative network when facing the significant domain shift between tasks.

Here are my detailed comments:
Catastrophic problem is the most severe problem in continual learning since when learning more and more new tasks, the classifier will forget what they learned before, which will be no longer an effective continual learning model. Considering that episodic memory will cost too much space, this work adopts the generative replay strategy where old representative data are generated by a generative model. Thus, at every time step, the model will receive data from every task so that its performance on old tasks will retain. However, if the differences between tasks are significant, the generator cannot reserve vacant neurons for new tasks or in other words, the generator will forget the old information from old tasks when overwritten by information from new tasks. Therefore, this work tries to tackle this problem by extending the output neurons of the generator to keep vacant neurons to retain receive new information. As far as I am concerned, this is the main contribution of this work.

Nevertheless, I think there are some deficiencies in this work.

First, this paper is not easy to follow. The main reason is that from the narration, I cannot figure out what is the idea or technique of other works and what is the contribution of this paper. For example, in Section 4.1, I am not sure the equation (3), (4), (5), (6) are the contributions of this paper or not since a large number of citations appear.

Second, the authors mention that to avoid storing previous data, they adopt generative replay and continuously enlarge the generator to tackle the significant domain shift between tasks. However, in this way, when more and more tasks come, the generator will become larger and larger. The storing problem still exists. Generative replay also brings the time complexity problem since it is time consuming to generate previous data. Thus, I suggest the authors could show the space and time comparisons with the baseline methods to show effectiveness of the proposed method.

Third, the datasets used in this paper are rather limited. Three datasets cannot make the experiments convincing. In addition, I observe that in Table 1, the proposed method does not outperform the Joint Training in SVHN with A_10. I hope the author could explain this phenomenon. Furthermore, I do not see legend in Figure 3 and thus I cannot figure out what the curves represent.

Fourth, there are some grammar mistakes and typos. For example, there are two "the" in the end of the third paragraph in Related Work. In the last paragraph in Related Work, "provide" should be "provides". In page 8, the double quotation marks of "short-term" are not correct.

Finally yet importantly, though a large number of works have been proposed to try to solve this problem especially the catastrophic forgetting, most of these works are heuristic and lack mathematical proof, and thus have no guarantee on new tasks or scenarios. The proposed method is also heuristic and lacks promising guarantee.

---

> ### Author Response · Authors · 2018-11-25
> **We thank the reviewer for their work. We address the comments of the reviewer as follows.**
>
> 1. We first want to point out the main contributions of the paper.
> First, we address the catastrophic forgetting problem in continual learning.  Thereby we introduce Dynamic Generative Memory (DGM) - an adversarially trainable generative network endowed with neuronal plasticity through efficient learning of sparse attention mask for layer activations. Hereby we extend the idea of [2] to generative networks. We highlight the differences to DGR [3] in the Sec. 2 of our work.
>
> 2. Equation (5) and (6) are taken from [2] one to one. Equations (3) and (4) are adopted from [2]: equation (3) describes the annealing of the parameter s, we anneal it globally over the course of epochs, whereas [2] anneal it for each epoch over the number of batches; equation (4) is a simplified version of the one used by [2].
>
> 3. To avoid confusion of the proposed method to utilize techniques of DGR[3] in order to prevent forgetting in the G, we kindly ask the reviewer to refer to our response (2) to the Reviewer 1.
>
> In the proposed work we adopt the generative replay not in order to avoid storing previous samples, but in order to prevent forgetting in the discriminator (which is used as a final classification model). Data synthesized by the generator is replayed for to the discriminator during the training of the subsequent tasks. There is no replay applied to the generator network. In order to avoid storing previous data, we utilize parameter level attention mechanism similar to HAT [2].
>
> Concerning the time comparison, there is no reason why our approach should be less time efficient then DGR based approaches [1, 3] as our method does not require retraining the generator from scratch at each time step.
>
> 4. Why our method does not outperform joint training on SVHN?
> Using generated samples accommodates for better performance then joint training is the case of tasks of relatively low complexity such as MNIST. Indeed, such a result has been shown in other works, e.g. [1]. As explained in Sec. 5.2, this can be attributed to a potentially higher diversity with a steady quality of the generated samples. Clearly, the performance of the classifier trained on the generated samples highly depends on the complexity of the task and quality of the generated samples. Thus, this effect can not be observed neither in the SVHN not the CIFAR10 benchmarks.
>
> 5. Grammar mistakes and typos.
> This will be fixed in the updated version of the paper.
>
> 6. No guarantee to work for any task or scenario.
> As pointed out by the reviewer and is true for many machine learning method, there is no guarantee that the proposed method will work for any task or scenario.
>
> [1] C. Wu, L. Herranz, X. Liu, Y. Wang, J. van de Weijer, and B. Raducanu. Memory Replay GANs: learning to generate images from new categories without forgetting. In Advances In Neural Information Processing Systems, 2018.
> [2] J. Serrà, D. Surís, M. Miron, and A. Karatzoglou. Overcoming catastrophic forgetting with hard attention to the task. CoRR, abs/1801.01423, 2018.  URL http://arxiv.org/abs/1801.01423.
> [3] H. Shin, J. K. Lee, J. Kim, and J. Kim.  Continual learning with deep generative replay.  In
> Advances in Neural Information Processing Systems, pages 2990–2999, 2017.

---

### Official Review · AnonReviewer3 · 2018-11-05
**Interesting combination of previous methods, but contributions are not clear and experiments need more rigor**

**Rating:** 4
**Confidence:** 5

**Review:**


The proposed method tackles class-incremental continual learning, where new categories are incrementally exposed to the network but a classifier across all categories must be learned. The proposed method seems to be essentially a combination of generative replay (e.g. Deep Generative Replay) with AC-GAN as the model and attention (HAT), along with a growing mechanism to support saturating capacity. Quantitative results are shown on MNIST and SVHN while some analysis is provided on CIFAR.

Pros

 + The method combines the existing works in a way that makes sense, specifically AC-GAN to support a single generator network with attention-based methods to prevent forgetting in the generator.

 + The method results in good performance, although see caveats below.

 + Analysis of the evolution of mask values over time is interesting.

Cons

 - The method is very confusingly presented and requires both knowledge of HAT as well as more than one reading to understand. The fact that HAT-like masks are used for a generative replay approach is clear, but the actual mechanism of "growing capacity" is not made clear at all especially in the beginning of the paper. Further the contributions are not clear at all, since a large part of the detailed approach/equations relate to the masking which was taken from previous work. The authors should on the claimed contributions. Is it a combination of DGR and HAT with some capacity expansion?

 - It is not clear whether pushing the catastrophic forgetting problem into the generator is the best approach. Clearly, replaying data accurately from all tasks will work well, but why is it harder to guard against the generative forgetting problem than the discriminative one?

 - The approach also seems to add a lot of complexity and heuristics/hyper-parameters. It also adds capacity and it is not at all made clear whether the comparison is fair since no analysis on number of parameters are shown.

 - Relatedly, better baselines should be used; for example, if the memory used by the generative model is merely put to storing randomly chosen instances from the tasks, how will the results compare? Clearly storing instances bypasses the forgetting problem completely (as memory size approaches the dataset size it turns into the joint problem) and it's not clear how many instances are really needed per task, especially for these simpler problems. As such, I find it surprising that simply storing instances would do as poorly as stated in this paper which says cannot provide enough diversity.

 It also seems strange to say that storing instances "violates the strictly incremental setup" while generative models do not. Obviously there is a tradeoff in terms of memory usage, privacy, performance, etc. but since none of these methods currently achieve the best across all of these there is no reason to rule out any of the methods. Otherwise you are just defining the problem in a way that excludes other simple approaches which work.

 - There are several methodological issues: Why are CIFAR results not shown in a table as is done for the other dataset? How many times were the experiments run and what were the variances? How many parameters are used (since capacity can increase?) It is for example not clear that the comparison to joint training is fair, when stating: "Interestingly, DGM outperforms joint training on the MNIST dataset using the same architecture. This suggests that the strictly incremental training methodology indeed forced the network to learn better generalizations compared to what it would learn given all the data." Doesn't DGM grow the capacity, and therefore this isn't that surprising? This is true throughout; as stated before it is not clear how many parameters and how much memory these methods need, which makes it impossible to compare.

 Some other minor issues in the writing includes:
   1) The introduction makes it seem the generative replay is new, without citing approaches such as DGR (which are cited in the related work). The initial narrative mixes prior works' contributions and this paper's contributions; the contributions of the paper itself should be made clear,

   2) Using the word "task" in describing "joint training" of the generative, discriminative, and classification networks is very confusing (since "task" is used for the continual learning description too,

   3) There is no legend for CIFAR; what do the colors represent?

   4) There are several typos/grammar issues e.g. "believed to occurs", "important parameters sections", "capacity that if efficiently allocated", etc.).

 In summary, the paper presents what seems like an effective strategy for continual learning, by combining some existing methods together, but does not make it precise what the contributions are and the methodology/analysis make it hard to determine if the comparisons are fair or not. More rigorous experiments and analysis is needed to make this a good ICLR paper.

---

> ### Author Response · Authors · 2018-11-19
> **Response to Reviewer #1 (part 2)**
>
>
> 4. Our approach has 2 important hyperparameters: scaling parameter s used for calculating binary mask from the embedding matrix as well as  λ_RU, that controls the size accuracy trade-off (see Sec. 4.1 “joint training”).  We add a table analyzing the sensitivity of the parameter λ_RU observing the expected behavior: higher values of λ_RU lead to a smaller model size, however, reduced G size is positively correlated with the final classification performance of D (smaller G -> lower accuracy of D).
> +---------+---------+-------+
> | λ_RU  | Acc.5 | Size |
> +---------+---------+-------+
> | 2E-06 | 98.16 | 660  |
> +---------+--------+--------+
> | 0.002 | 98.22 | 638  |
> +---------+--------+--------+
> | 0.2     | 98.02 | 598  |
> +---------+--------+--------+
> | 0.75   | 97.36 | 577  |
> +---------+--------+--------+
> | 2        | 86.82 | 522  |
> +---------+--------+--------+
>
> 5. We use the baseline presented by [1], that tackles identical scenario. To our knowledge [1] provides the state of the art performance in "strict" class incremental setup without using real samples.
>
>  We consider a joint training (JT, classical training) of the discriminator as the upper performance bound. Joint training features a setup in which the discriminator is trained on ALL real samples of the previous tasks. The reviewer proposes to simulate information loss and use a random subset of real samples to train the upper bound model. However, this would certainly give a worse performance than when using all real samples. We, therefore, think that used JT upper bound is appropriate.
>
> Furthermore, using generated samples accommodates for better performance than simply storing instances only in case of tasks of relatively low complexity such as MNIST. Indeed, such a result has been shown in other works, e.g. [1]. As explained in Sec. 5.2, this can be attributed to a potentially higher diversity with steady quality of the generated samples. Clearly, the performance of the classifier trained on the generated samples highly depends on the complexity of the task and quality of the generated samples. Thus, this effect can be observed neither in the SVHN nor the CIFAR10 benchmarks.
>
> 6. The CIFAR results will be provided in the Tab. 1 alongside with other datasets in the next version.
>
> To ensure a fair comparison with the benchmark methods that do not use any network expansion strategy for the generator (e.g. [1,6]), we initialize our G to be approximately 50% of the size of the G used in the benchmarks. Also a study on network growth dynamics is provided in Fig. 5 (Sec. 5.3), showcasing a lower network capacity than the worst case scenario. Growing the generator is an essential part of our method that addresses the scalability problem in continual learning, e.g. with always growing amount of data model’s capacity will be exhausted at a certain point. Noteworthy, the discriminator is not affected by the proposed dynamic network expansion mechanism and features the same architecture as in the benchmark methods.
>
> We believe the comparison to the joint training is fair because DGM only grows the capacity of the generator. In the discriminator, only the last classification layer is expanded with the growing model’s output space as new classes are added. Thus, for k-th task we compare the accuracy of a discriminator with identical architecture trained on real samples of all k tasks (JT) with one trained on DGM-synthesized samples of k-1 tasks+reals of k-th tasks. Thus DGM’s discriminator has no advantages over the joint training generator.
>
> 8. Finally, we will address typos, writing and presentation issues in the updated version of the paper.
>
>
> [1] C. Wu, L. Herranz, X. Liu, Y. Wang, J. van de Weijer, and B. Raducanu. Memory Replay GANs: learning to generate images from new categories without forgetting. In Advances In Neural Information Processing Systems, 2018.
>
> [2] J. Serrà, D. Surís, M. Miron, and A. Karatzoglou. Overcoming catastrophic forgetting with hard attention to the task. CoRR, abs/1801.01423, 2018.  URL http://arxiv.org/abs/1801.01423.
>
> [3] H. Shin, J. K. Lee, J. Kim, and J. Kim.  Continual learning with deep generative replay.  In
> Advances in Neural Information Processing Systems, pages 2990–2999, 2017.
>
> [4] S. Rebuffi, A. Kolesnikov, and C. H. Lampert. icarl: Incremental classifier and representation
> learning.CoRR, abs/1611.07725, 2016. URL http://arxiv.org/abs/1611.07725.
>
> [5] A. Chaudhry, P. K. Dokania, T. Ajanthan, and P. H. S. Torr. Riemannian walk for incremental learning: Understanding forgetting and intransigence. CoRR, abs/1801.10112, 2018. URL http://arxiv.org/abs/1801.10112.
>
> [6] Hanul Shin, Jung Kwon Lee, Jaehong Kim, and Jiwon Kim. Continual learning with deep generative replay. In Advances in Neural Information Processing Systems, pp. 2990–2999, 2017.

---

> > ### Comment · AnonReviewer3 · 2018-11-23
> > **Experimental methodology**
> >
> >
> >  Thanks again for the responses.
> >
> > > Furthermore, using generated samples accommodates for better performance than simply storing instances
> > > only in case of tasks of relatively low complexity such as MNIST.
> >
> >   Sure, but 1) this makes it not surprising like it's presented in the paper (there are a large number of papers that essentially use generative models as data augmentation, and you could do this for your joint training methods as well) and 2) I was saying that I'm surprised that methods such as iCARL or simply replaying a small number of examples wouldn't do well on these tasks.
> >
> > > The CIFAR results will be provided in the Tab. 1 alongside with other datasets in the next version.
> >
> >   If you have these available, can you post them on openreview?
> >
> >   In terms of experimental methodology, I don't believe growing the generator is fair, or at least it brings in other competitors that do the same. Specifically, replay methods that use real samples typically reduce the number of samples per task as the number of tasks grow, specifically to keep the memory constant. Here, while you are not growing the discriminator, you are still growing the amount of memory you use. Again, a simple baseline would be to take the same amount of memory your method uses (including expansion) and replay those examples during training. In all of these comparisons, a table is needed that shows exactly how much memory is used for all of the baselines/competitors that use replay, and how much memory your method uses. Note the other reviewer asked for the same, and included time complexity as well.
> >
> >   Further, if you are going to use capacity expansion there are a number of methods that aren't cited in your work, including progressive networks [1,2] the latter of which uses distillation as a mechanism to avoid large-scale growth in the networks.
> >
> >    This paper and results does have promise, but given that it essentially uses HAT for the generative model, this is largely an empirical paper. As such, there should be precise experiments that make it much easier to discern the advantage of the method over both state of art as well as much simpler baselines.
> >
> > [1] Progressive Neural Networks, Andrei A. Rusu, Neil C. Rabinowitz, Guillaume Desjardins, Hubert Soyer, James Kirkpatrick, Koray Kavukcuoglu, Razvan Pascanu, Raia Hadsell, https://arxiv.org/abs/1606.04671
> >
> > [2] Progress & Compress: A scalable framework for continual learning, Jonathan Schwarz, Jelena Luketina, Wojciech M. Czarnecki, Agnieszka Grabska-Barwinska, Yee Whye Teh, Razvan Pascanu, Raia Hadsell, https://arxiv.org/abs/1805.06370

---

> > > ### Author Response · Authors · 2018-11-26
> > > **Thank you for your feedback.**
> > >
> > > We updated the paper with the CIFAR results as well as cite the mentioned papers on capacity growth.
> > >
> > > Considering the comparison to Progressive Networks:
> > > Similarly to Progressive Neural Networks [1] and its evolution [2] our method addresses the challenge of knowledge transfer by ensuring the reusability of parameters across the tasks. Our method does it naturally since it only keeps a single network for long and short-term memory with different neurons assigned to different memory types. Using binary masking allows keeping both memory types in a single network without forgetting. DGM neither require keeping a pool of networks (columns) used for previous tasks (as in [1]) nor utilizing separate long and short-term memory networks (as in [2]).
> > >
> > > Overall, we thank the reviewer again for the constructive feedback, which we will consider in our future work.
> > >
> > > [1] Progressive Neural Networks, Andrei A. Rusu, Neil C. Rabinowitz, Guillaume Desjardins, Hubert Soyer, James Kirkpatrick, Koray Kavukcuoglu, Razvan Pascanu, Raia Hadsell, https://arxiv.org/abs/1606.04671
> > > [2] Progress & Compress: A scalable framework for continual learning, Jonathan Schwarz, Jelena Luketina, Wojciech M. Czarnecki, Agnieszka Grabska-Barwinska, Yee Whye Teh, Razvan Pascanu, Raia Hadsell, https://arxiv.org/abs/1805.06370

---

> ### Author Response · Authors · 2018-11-19
> **Response to Reviewer #1 (part 1)**
>
> We thank the reviewer for their constructive comments. We address them as follows.
>
> 1. We first would like to point out the contributions of our work.
>
> First, we address the catastrophic forgetting problem in continual learning.  Thereby we introduce Dynamic Generative Memory (DGM) - an adversarially trainable generative network endowed with neuronal plasticity through efficient learning of sparse attention mask for layer activations. Hereby we extend the idea of HAT[2] to generative networks.
>
> Secondly, we address the scalability problem in continual learning. To ensure sufficient model capacity to accommodate for new tasks, we propose an adaptive network expansion mechanism in which newly added capacity is derived from the learnable neuron masks.
>
>
> 2. We further we would like to clarify a possible confusion of the proposed method to be a combination of Deep Generative Replay (DGR)[6] and HAT[2].
>
> As pointed out in the Sec. 2 of our work, Deep Generative Replay (DGR) tries to prevent forgetting in the generator by retraining it from scratch every time a new data chunk becomes available. Thus, in DGR the generator would lose information at each replay step since the quality of generated samples highly depends on the quality of samples generated by the prior generator causing "semantic drift". This contrasts our method, which effectively retains the knowledge in the generator using HAT like neuron masking and only loses information through “natural” forgetting.  This allows us to use “complete” learned representation during learning and inference of the subsequent tasks as well as speed up the training (no replay of G is involved).
>
> 3. We are not simply shifting the forgetting problem into G.
>
> Our work tackles the problem of class incremental learning.  As opposed to task-incremental setup and shown in previous work, e.g. [3,4,5], models in class incremental setup (with single-head architecture) require a replay of previously seen categories when learning new ones. The reason for using G is not having access to samples of previous classes in the “strict” incremental setup and using generated samples instead. As pointed out in our work, restricting storage of real samples represents a more realistic setup, since in real-world applications such an “episodic memory” with real samples is often impossible due to memory and privacy restrictions.

---

> > ### Comment · AnonReviewer3 · 2018-11-23
> > **Thank you for the response**
> >
> >
> >  I appreciate the authors' detailed response.
> >
> >   In terms of contribution, it would be great to make this much more clear in the revision. However, you seem to agree that essentially it is to extend HAT to the generative network in addition to adding (a very simple) capacity expansion, neither of which adds a great amount of novelty or advances our understanding of continual learning. For example, as far as I can tell most of the technical description in section 4 is for HAT.
> >
> >   I also still do not understand why the method is claimed to be in such stark contrast to the previous work. When you say that your method only loses information through "natural forgetting" what does this mean precisely? What does "'complete' learned representation" mean? These are vague terms and should be precisely defined. To me both of these are again just achieved by using HAT.
> >
> >   Finally, I am still not convinced why a "strict" incremental setting rules out the storage of real samples. If storing a number of examples equivalent to the amount of memory used by your method achieves better performance (of course, equalizing for the capacity growth you have), why is that an issue? The only reason I agree with might be privacy, but that can be addressed through other privacy methods (e.g. random perturbation).
> >
> >   Overall, I think generative models are great and can be useful to addressing catastrophic forgetting, but such methods have have other limitations (complexity of training, etc.) and should be compared to simpler replay baselines. Further, just the idea of using HAT for a generative model plus capacity expansion seems to me not a significant enough contribution. Given the additional concerns about methodology (see next comment), I do not believe these comments sufficiently address all of the concerns to warrant acceptance.

---

> > > ### Author Response · Authors · 2018-11-26
> > > **Why “strict” is not only privacy.**
> > >
> > > We thank the reviewer again for his/her extensive response.
> > >
> > > We believe, the just storing real samples of previous classes does not comply with the fundamental vision of how a continually trainable system should work (e.g. compared to natural intelligence). Further, the challenge of scalability in continual learning cannot be addressed by simply storing real samples (at least not in large-scale context). “Strictness” is an increasingly important issue in the literature and has been addressed by other works such as [ 1,2,3 ]. We, therefore, stick to “strictness” requirement and prohibit storing real samples which naturally leads to using the generative memory.
> > >
> > > As opposed to DGR [1] based approaches, DGM replays a 'complete' learned representation of previous tasks - meaning no information is lost due to continuous retraining of the G on samples generated by the previous generator.
> > >
> > >
> > > [1] Hanul Shin, Jung Kwon Lee, Jaehong Kim, and Jiwon Kim. Continual learning with deep generative replay. In Advances in Neural Information Processing Systems, pp. 2990–2999, 2017.
> > > [2] C. Wu, L. Herranz, X. Liu, Y. Wang, J. van de Weijer, and B. Raducanu. Memory Replay GANs: learning to generate images from new categories without forgetting. In Advances In Neural Information Processing Systems, 2018.
> > > [3] Seff, Ari, et al. "Continual learning in generative adversarial nets." arXiv preprint arXiv:1705.08395(2017).

---

### Meta-Review · Area_Chair1 · 2018-12-15

**Confidence:** 5
**Recommendation:** Reject

**Metareview:**

The authors propose to tackle the problem of catastrophic forgetting in continual learning by adopting the generative replay strategy with the generator network as an extendable memory module.

While acknowledging that the proposed model is potentially useful, the reviewers raised several important concerns that were viewed by AC as critical issues:
(1) poor presentation clarity of the manuscript and incremental technical contribution in light of prior work by Serra et al. (2018); (2) rigorous experiments and in-depth analysis of the baseline models in terms of accuracy, number of parameters, memory demand and model complexity would significantly strengthen the evaluation – see R1’s and R3’s suggestions how to improve; (3) simple strategies such as storing a number of examples and memory replay should not be neglected and evaluated to assess the scope of the contribution.
Additionally R1 raised a concern that preventing the generator from forgetting should be supported by an ablation study on both, the discriminator and the generator, abilities to remember and to forget.

R1 and R3 provided very detailed and constructive reviews, as acknowledged by the authors. R2 expressed similar concerns about time/memory comparison of different methods, but his/her brief review did not have a substantial impact on the decision.

AC suggests in its current state the manuscript is not ready for a publication. We hope the reviews are useful for improving and revising the paper.